# Simple models including energy and spike constraints reproduce complex activity patterns and metabolic disruptions

**Tanguy Fardet**[1,2]*, **Anna Levina**[1,2]

**1** University of Tübingen, Tübingen, Germany, **2** Max Planck Institute for Biological Cybernetics, Tübingen, Germany

* tanguy.fardet@ens-lyon.org

## Abstract

In this work, we introduce new phenomenological neuronal models (*e*LIF and mAdExp) that account for energy supply and demand in the cell as well as the inactivation of spike generation how these interact with subthreshold and spiking dynamics. Including these constraints, the new models reproduce a broad range of biologically-relevant behaviors that are identified to be crucial in many neurological disorders, but were not captured by commonly used phenomenological models. Because of their low dimensionality *e*LIF and mAdExp open the possibility of future large-scale simulations for more realistic studies of brain circuits involved in neuronal disorders. The new models enable both more accurate modeling and the possibility to study energy-associated disorders over the whole time-course of disease progression instead of only comparing the initially healthy status with the final diseased state. These models, therefore, provide new theoretical and computational methods to assess the opportunities of early diagnostics and the potential of energy-centered approaches to improve therapies.

**Data Availability Statement:** Data used in the paper come from the Allen Institute for Brain Science (2015). Allen Cell Types Database. Available from: http://celltypes.brain-map.org (cells 566978098 and 570896413). Code for models

## Author summary

Neurons, even "at rest", require a constant supply of energy to function. They cannot sustain high-frequency activity over long periods because of regulatory mechanisms, such as adaptation or sodium channels inactivation, and metabolic limitations. These limitations are especially severe in many neuronal disorders, where energy can become insufficient and make the neuronal response change drastically, leading to increased burstiness, network oscillations, or seizures. Capturing such behaviors and impact of energy constraints on them is an essential prerequisite to study disorders such as Parkinson's disease and epilepsy. However, energy and spiking constraints are not present in any of the standard neuronal models used in computational neuroscience. Here we introduce models that provide a simple and scalable way to account for these features, enabling large-scale theoretical and computational studies of neurological disorders and activity patterns that could not be captured by previously used models. These models provide a way to study

implementation is freely available at https://github.com/Silmathoron/elif-madexp and on ModelDB.

**Funding:** TF was awarded a Humboldt Research Fellowship for Postdoctoral Researchers and AL was awarded a a Sofja Kovalevskaja Award from the Alexander von Humboldt Foundation: https://www.humboldt-foundation.de. The funders had no role in study design, data collection and analysis, decision to publish, or preparation of the manuscript.

**Competing interests:** The authors have declared that no competing interests exist.

energy-associated disorders over the whole time-course of disease progression, and they enable a better assessment of energy-centered approaches to improve therapies.

## 1 Introduction

Brain metabolism, even in its resting state, constitutes a major source of energy consumption in mammalian species. Indeed, cells—and especially excitable cells such as neurons—undergo constant ion fluxes both along and against the concentration and electric gradients. To move ions against these gradients, an active mechanism is required, which consumes energy in the form of ATP. In cells, this work is mostly associated with the sodium-potassium pump (Na/K pump or NKP) which moves 3 sodium ions out of the cell in exchange for 2 potassium ions moving in for every hydrolyzed ATP molecule, thus creating a net electric current [1]. As a result, Na/K pump is responsible for roughly 75% of the total energy consumption in neurons [2], which arguably makes it one of the most important players in the cell: its action makes the energy from the hydrolysis of ATP available to most other processes [3], allowing changes in the membrane potential, regulation of the volume, or transport of nutrients inside the cell. Thus the energy level, through the Na/K pump activity, modulates neuronal response and directly influences information processing [4].

Though the Na/K pump has been thoroughly researched in the past decades [1, 3], surprisingly few neuronal models include the pump and its electrogenic properties [5–7] and even fewer account for its underlying energy substrate [8, 9]. A probable reason for this fact comes from the significant focus of theoretical studies on cortical areas that generally display sparse activity. Such conditions put little or no metabolic stress on the neurons and thus limit the influence of the Na/K pump and energetic constraints on the dynamics. However, the story changes drastically when energy-intensive behaviors such as bursting or fast pacemaking dynamics are considered, or when studying neuronal disorders. Indeed, both situations can place neurons under significant metabolic stress and induce fluctuation in the metabolite and ion concentrations which, from NKP-driven coupling between metabolism and activity, can then lead to major changes in the neuronal dynamics.

Outside of neuroscience, the influence of Na/K pump and energy consumption on activity and disorders were investigated in the context of the cardiac electrophysiology [10–12]. However, awareness is now raising in the neuroscience community, including its most theoretically-oriented members, as an increasing number of publications start to stress the critical influence of mitochondria [13, 14] and Na/K pump [15] and the intricate feedback loops between activity and energetics. Some well-known works on energetics in computational neuroscience include the energy budgets from [16] and [2], as well as studies related to the link between action potential shape and ATP consumption [17, 18]. Yet, these studies deal with general budgets from the point of view of optimality theory and do not describe the local interactions between energy levels and spike initiation.

The interactions between energetics and neuronal activity are most visible in neuronal disorders such as epilepsy [19–21], Alzheimer [22], or Parkinson's disease [23, 24]. It is therefore in the context of neuronal diseases that one can find the few studies that really focused on these interactions [8, 9, 25, 26]. Unfortunately, because such studies are still scarce and the associated modeling frameworks are still limited, computational studies of neuronal disorders currently suffer from at least one of the following issues: a) they do not account for energetic constraints, b) the models do not reproduce important features of the relevant neuronal

behaviors, or c) the size of the simulated networks is extremely small (notably due to the use of complex conductance-based models).

Here we present new models to help tackle these issues through theoretical descriptions of neuronal dynamics that a) account for energy levels and their influence on neuronal behavior, b) are able to reproduce most relevant neuronal dynamics in the context of disorders such as seizures or Parkinson's disease, and c) can be used in simulation of networks up to several million neurons.

## 2 Methods

In the following, we describe the implementation of the new models. We discuss the biological mechanisms that gave rise to the variables and equations in our models and list the associated properties that an energetic model should satisfy. Two major biological components considered in the models are the pumps that degrade ATP into ADP to maintain ion gradients (most notably the Na/K and calcium pumps), and ATP-gated potassium (K-ATP) channels that open or close depending on the ATP/ADP ratio. When ATP concentration or the ATP/ADP ratio decreases, the pump's effectiveness decreases, resulting in a rise of sodium concentration, thus increasing membrane potential and sometimes excitability. Conversely, a decrease in the ATP/ADP ratio tends to open K-ATP channels, allowing potassium to flow out of the cell and decrease membrane potential and excitability. These mechanisms directly or indirectly influence a neuron's excitability and its ability to generate action potentials. Depending on their relative importance, a neuron can, therefore, end up in a depolarized or hyperpolarized state when energy levels decrease.

Based on these main mechanisms, the models were design to meet several conditions that can be split into 1) behavioral requirements, associated to the type of responses and biological situations that the models can account for, and 2) practical constraints associated to the computational cost and theoretical complexity of the model. As energetic constraints are especially relevant for behaviors associated with diseased or hypoxic state, we designed our models so that they would be able to provide meaningful behaviors in such conditions.

Regarding behavioral requirements, we took care of reproducing the effects ATP/ADP changes on pumps and K-ATP channels so that the models could account for three major observations:

- as mitochondrial health or metabolic resources decrease (e.g. during hypoxia), the excitability and resting potential of the neuron can increase [25, 27], notably due to insufficient activity of the Na/K pump,

- decrease in metabolic resources is also associated with an increase in calcium levels [27] due to insufficient activity of the pumps,

- during seizures, or when submitted to excessive excitation, neurons undergo depolarization blocks characterized by "superthreshold" membrane potential without spike emission [28] that is caused by sodium-channel inactivation as the Na/K pump cannot move sodium out quickly enough.

In addition, the specific form of the equation was chosen to allow two specific behaviors to be switched on or off depending on the parameters used:

- neuronal bistability, observed in several brain regions [29, 30], is involved in important mechanisms such as up-and-down states and could also explain discontinuities in the progression of neurodegenerative diseases [31],

- adaptation currents and bursting or rebound activities that are major players in neuronal disorders [32, 33].

Our central goal is to develop models that do not only reproduce important behaviors, but also allow for large-scale event-based simulations. To achieve this, the computational cost and complexity of the models should be minimal. Thus, we decided to use hybrid models based on the integrate-and-fire paradigm.

We established that models including an adaptation current, such as the Quadratic Integrate-and-Fire and the AdExp neurons [34, 35], were able to provide most of the required dynamics such as bursting and rebound activity [36, 37]. The missing requirements—depolarization block and bistability—as well as the inclusion of metabolic resources would thus come from the addition of dynamic resource availability (broadly called energy in the following), as shown on Fig 1. This purpose of this variable is to represent the ATP/ADP ratio in biological neurons, though the phenomenological nature of the models implies that there are limits to this analogy.

For applications where bursting behavior and adaptation do not play an important role, a simple model that accounts only for energy dynamics is provided: the *e*LIF neuron. It introduces energy dynamics as an addition to the simpler leaky integrate-and-fire (LIF) model and enables us to analyze the consequences of these constrains in a more straightforward and visual manner. The behavior of this model can also be fully investigated analytically compared to the 3-dimensional system that arises in a second time when both energy and adaptation dynamics are considered. This second model, called mAdExp, is built upon the AdExp equations and

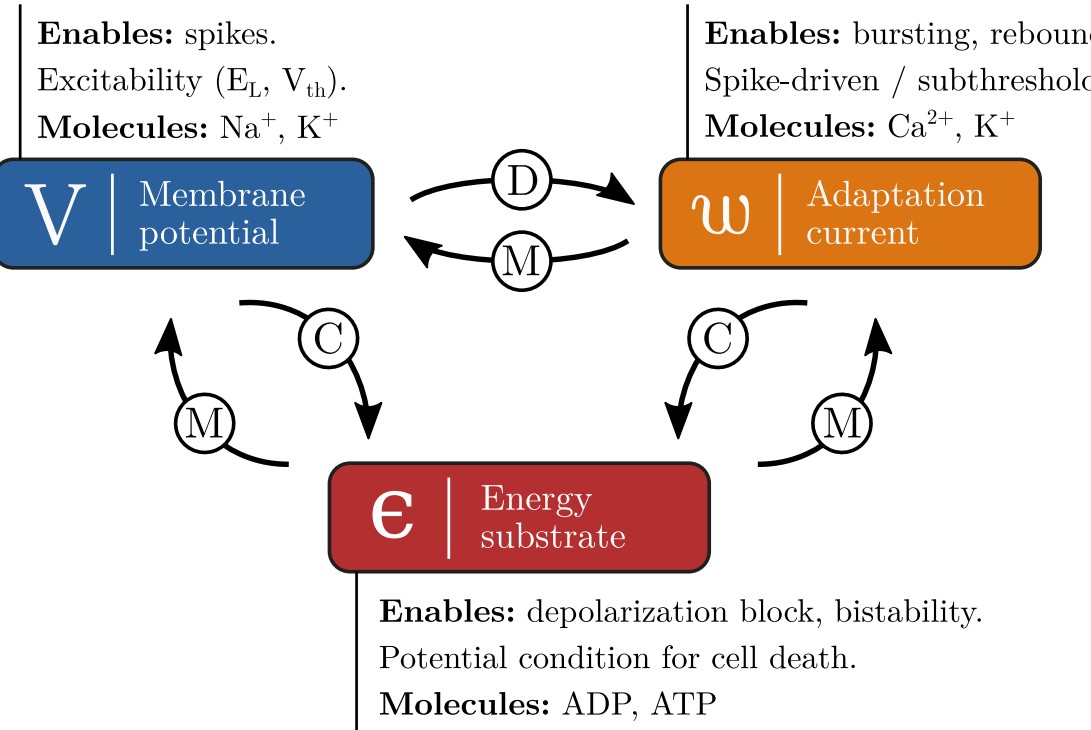

**Fig 1. Variables and interactions that must be present in the models to capture all relevant behaviors, the main molecules associated to each of the variables are also displayed.** The type of interaction is marked on the arrow. For instance, *w* modulates (M) *V* as it influences the intrinsic dynamics of *V* but does not usually cause it directly. On the other hand, as changes in the membrane potential are the main cause of variations in *w*, *V* is said to drive (D) *w*. Eventually, all mechanisms consume (C) energy.

cam reproduce all desired behaviors. Though analytical analysis of this model can prove challenging, most of its dynamics can be understood from the complementary analyses of the $e$LIF and AdExp models.

## 2.1 Introducing energy: The $e$LIF model

The first proposed model is a straightforward modification of the standard Leaky Integrate-and-Fire (LIF) model [38]. In order to provide an intuitive and analytically tractable implementation that would illustrate the consequences of energy dynamics and the constraints it places on spike-emission, we developed a two-dimensional dynamical system describing the evolution of a) the membrane potential $V$ of a point neuron and b) the available amount of energy $\epsilon$ that the neuron can access. To make the equations more readable and the parameters easy to interpret, the model is displayed using three equations; however, it can be easily simplified to a system of two equations only.

$$\text{if}\quad V < V_{th}\quad\text{or}\quad \epsilon < \epsilon_c \begin{cases} C_m \dot{V} &= g_L(E_L - V) + I_{syn} + I_e \\ \tau_e \dot{\epsilon} &= \left(1 - \dfrac{\epsilon}{\alpha\epsilon_0}\right)^3 - \dfrac{V - E_f}{E_d - E_f} \\ E_L &= E_0 + (E_u - E_0)\left(1 - \dfrac{\epsilon}{\epsilon_0}\right) \end{cases} \text{else} \begin{cases} V &\leftarrow V_r \\ \epsilon &\leftarrow \epsilon - \delta \end{cases} \quad (1)$$

As in other standard integrate-and-fire models, the neuron possesses a leak potential $E_L$, a membrane capacitance $C_m$, and a leak conductance $g_L$, the combination of the last two defining the membrane timescale $\tau_m = C_m/g_L$. Input from other neurons are represented by $I_{syn}$ while external input currents are associated to $I_e$. When either of these inputs brings the neuron above its threshold potential $V_{th}$, provided that there is enough energy ($\epsilon > \epsilon_c$) a spike is emitted and the voltage is instantaneously reset to $V_r$.

The available energy $\epsilon$ is introduced as a proxy for the ATP/ADP ratio in biological neurons. Its value varies with a typical timescale $\tau_e$ and is regulated by an interplay of the energy production (which tries to maintain it close to the typical energy value defined by the energetic health $\alpha\epsilon_0$) and two consumption mechanisms. The production term reflects mostly the oxidative phosphorylation performed in mitochondria [39] that enables a tight regulation of ATP levels in the cell.

The first consumption mechanism is associated with the fluctuations of the membrane potential and accounts for the ATP consumed by the Na/K pump to maintain ion homeostasis [3]. Since there is no available information about the functional form of the relationship between membrane potential and energy consumption, we have almost no constraints on the choice of the functional class. We selected a function allowing for a wide range of behaviors as observed in experiments while remaining as simple as possible: a 3rd order polynomial (see Fig 2, red line). Indeed, this is the simplest nonlinearity that can, depending on parameter values, either lead to a behavior that is qualitatively equivalent to a linear relationship or to the presence of a bistability, making it possible to asses the influence of bistable states on neurons' and network dynamics. The parameters defining the shape of the nullcline are: the flex potential $E_f$ (that corresponds to the inflection point, or *flex*, of the curve) and the energy-depletion potential $E_d$, that is a potential at which $\epsilon$-nullcline crosses the x-axis—$E_d$ thus corresponds to the lowest voltage-clamp potential that will lead to complete energy depletion and therefore neuronal death.

The second source of energy consumption is the energetic cost $\delta$ of the spike generation mechanisms. Though the biological reason for this energy consumption is the same as the

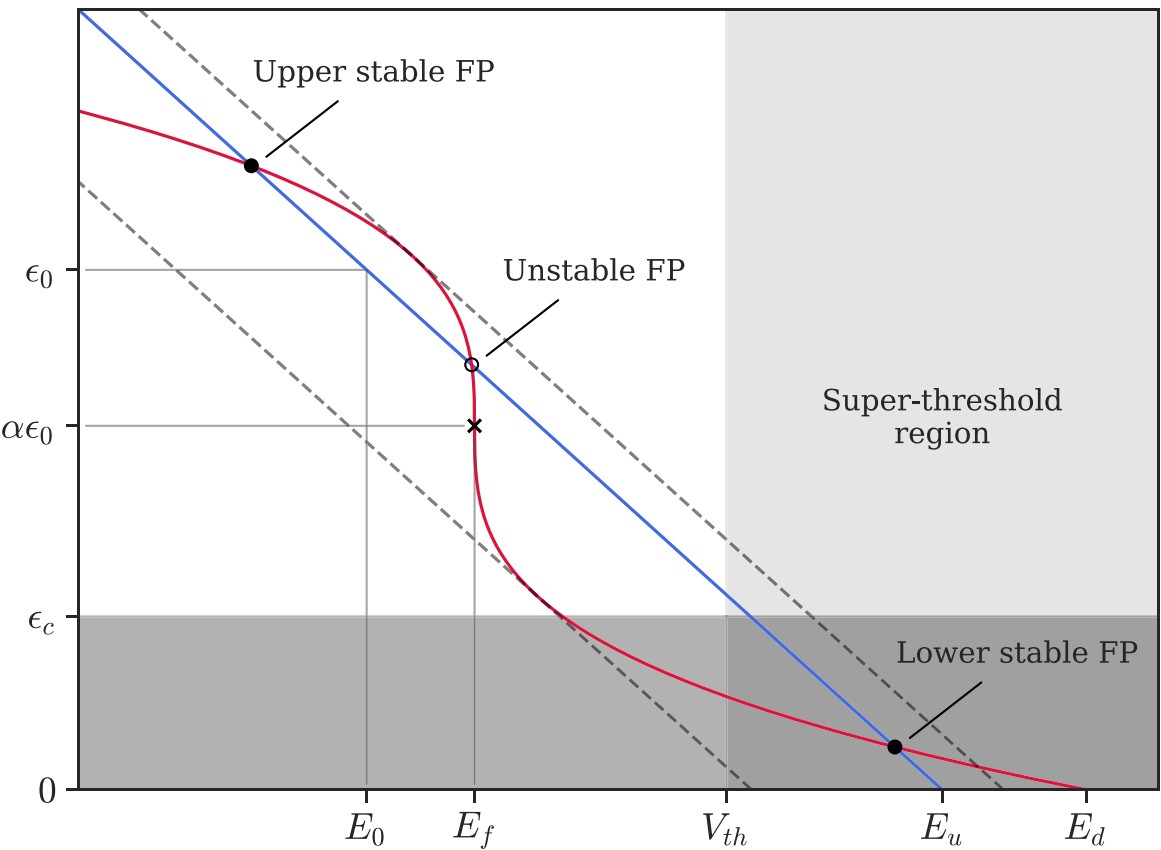

**Fig 2. Phase space of the *e*LIF model in bistable parameter regime.** *V*-nullcline is given by the blue line, $\epsilon$-nullcline by the red curve. Fixed points (FPs) are shown by the circles (filled for stable and empty for unstable) and the cross marks the inflection point of the $\epsilon$-nullcline. Dashed lines represent the shifts in the *V*-nullcline which lead to the disappearance of the unstable fixed point and of one of the stable fixed points (saddle-node bifurcation via the external current $I_e$). The super-threshold region, where spikes are elicited upon entrance, is marked by the light grey shading; the energy-limiting region ($\epsilon < \epsilon_c$) is marked by the grey shading and overlaps with the super-threshold region in the dark grey area, where energy limitations prevent spiking though the neuron is above threshold.

first term (ionic transfer by the Na/K pump), a separate term is necessary because of the reset mechanism of integrate-and-fire neurons: in such models, part of the spike duration is compressed into an instantaneous jump; $\delta$ thus accounts for the energy consumed during this compressed period. The normal energy level that the neuron is able to maintain depends on its "energetic health" described by the $\alpha$ parameter: a healthy neuron would have a value of $\alpha$ equal to one, while diseased neuron would see their $\alpha$ parameter decrease towards zero.

Contrary to most previous models, the leak potential is not constant, as it depends on the energy level of the neuron. The steady-state value $E_L$ of the membrane potential thus varies linearly, starting from $E_u$ when the energy is zero and decreasing as $\epsilon$ increases, crossing the potential $E_0$ for $\epsilon = \epsilon_0$ (see Fig 2) for details). Biologically, this account for the fact that a decrease in energy availability inhibits the function of the Na/K pump, leading to sodium accumulation inside the cell and thus to a depolarization.

The behavior of the standard LIF is recovered when $E_u = E_0$ and $\delta = 0$.

## 2.2 Adaptation and bursting: mAdExp model

In order to model the whole range of biologically-relevant behaviors that can be observed in neuronal disorders such as epilepsy or Parkinson's disease, it is necessary to include a modulatory mechanism to account for cellular and spike-driven adaptation. This second dynamical system keeps the basic properties introduced in the *e*LIF model and extends them to accommodate the cellular adaptation and spike initiation mechanisms of the adaptive Exponential Integrate-and-Fire model (aEIF or AdExp) by [35]. This leads to a 3D model with three dynamical state variables which are the membrane potential $V$, the energy level $\epsilon$ (as for the *e*LIF model), and an adaptation current $w$:

$$\text{if } V < V_{peak} \begin{cases} C_m \dot{V} &=& g_L(E_L - V) + g_L \Delta_T \dfrac{\epsilon - \epsilon_c}{\epsilon_0} \exp\left(\dfrac{V - V_{th}}{\Delta_T}\right) - w + I_{syn} + I_e \\[2ex] \tau_e \dot{\epsilon} &=& \left(1 - \dfrac{\epsilon}{\alpha \epsilon_0}\right)^3 - \dfrac{V - E_f}{E_d - E_f} - \dfrac{w}{\gamma} \\[2ex] \tau_w \dot{w} &=& a(V - E_L) - w + \dfrac{\epsilon_c}{\epsilon_c + 2\epsilon} I_{KATP} \\[2ex] E_L &=& E_0 + (E_u - E_0)\left(1 - \dfrac{\epsilon}{\epsilon_0}\right) \end{cases} \text{else} \begin{cases} V &\leftarrow& V_r \\[1ex] w &\leftarrow& w + b \\[1ex] \epsilon &\leftarrow& \epsilon - \delta \end{cases} \quad (2)$$

Compared to the *e*LIF implementation, the presence of the spike initiation mechanism through the exponential function removes the necessity of a hard threshold for spike prevention due to energy limitation: the $(\epsilon - \epsilon_c)/\epsilon_0$ factor suppresses the exponential divergence as soon as the amount of available energy goes below $\epsilon_c$.

The dynamics of the $\epsilon$ variable remains mostly unchanged except for the addition of a new consumption term associated with the adaptation current $w$: biologically $\gamma^{-1}$ corresponds to the energetic cost of bringing back the potassium ions which exited the cell (through calcium-gated potassium channels) per pA unit of the adaptation current. The model thus clearly separates the contributions of the energy ($\epsilon$) and of the calcium-gated adaptation ($w$).

Compared to the original AdExp model, the $w$ dynamics includes an additional term, $\frac{\epsilon_c}{\epsilon_c + 2\epsilon} I_{KATP}$, to account for ATP-sensitive potassium channels that trigger potassium outflow when the ATP/ADP ratio becomes small, with a typical activation-threshold depending on the ADP/ATP ratio [40]. $I_{KATP}$ is thus the maximum current at zero energy. Because of the numerous calcium exchangers in neuronal cells [41, 42], the term responsible for the exponential decay of the adaptation current with timescale $\tau_w$ is considered to be energy-independent. Thus, only $E_L$ and K-ATP induce energy-dependent changes in the adaptation current.

## 2.3 Numerical implementations

Implementations of the models are available for three major simulation platforms: NEST [43], through the NESTML language [44], BRIAN [45], and NEURON [46]. Models are available on ModelDB and on GitHub (https://github.com/Silmathoron/elif-madexp), together with code to reproduce the figures. Networks were generated using NNGT 2.0 [47] and simulated using NEST 2.20 [43]. Benchmarks have been performed with NEST and can be found in S1 Table.

## 2.4 Fitting procedure

To reproduce experimental recordings, we could set some of the model parameters directly from the data. The rest had to be manually adjusted. The following parameters can be informed from the data: a) $E_L$ was obtained by measuring the median resting value b) the

membrane timescale $\tau_m$ was measured from the initial slope of the membrane dynamics in response to hyperpolarizing currents c) the sum $g_L + a$ was obtained through a linear regression from the difference between resting $E_L$ and steady-state $E_{ss}$ potentials in response to depolarizing currents as $\Delta V = E_{ss} - E_L = \frac{I}{a+g_L}$. These properties were used to constrain the following parameters: $C_m, g_L, a, E_L, E_0, E_u$. All other parameters were then manually adjusted to minimize the discrepancy between subthreshold dynamics, number and time of spikes. Further research would be necessary to find how to automate this procedure using a proper distance function in optimization toolboxes.

## 3 Results

The new $e$LIF and mAdExp models enable us to obtain a variety of new dynamics such as rebound spiking, depolarization block, cellular bistability and up-and-down states, as well as biologically relevant transitions from a healthy to a diseased state.

For hybrid models, most of the neuronal dynamics can be understood through two main concepts: a) fixed points (FPs), which are equilibrium states of the model, and b) bifurcations, which are sudden changes in the number or stability of the fixed points, and which make the neuron change its behavior, for instance from resting to spiking.

This section details the aforementioned behaviors and their mechanistic origins through the theory of dynamical systems, using fixed points and bifurcations.

### 3.1 Behaviors and bifurcations of the $e$LIF model

The $e$LIF model, like the integrate-and-fire (LIF) neuron, has only two dynamical states: quiescent or active (spiking). Due to the energetic constraints, the model has two possible quiescent states which are the "normal" resting state, with a membrane potential located below threshold, and a super-threshold state where depleted energy levels prevent spike emission. The finite energy resources also imply that, contrary to the LIF neuron, the active state can be transient, as the neuron transits from its resting state to a quiescent, super-threshold state through an active period.

In the language of dynamical systems, the quiescent states are associated to FPs inside the continuous region (if either $V_{FP} < V_{th}$ or $\epsilon_{FP} < \epsilon_c$), whereas the active state is associated to the absence of a stable FP that can be accessed continuously in the region of phase-space where the neuron lies—see Fig 3.

We will focus here on the situation that is most relevant for the study of neuronal disorders, i.e. the case where $E_u > E_0$, meaning that decrease in energy levels leads to increase in membrane potential. This situation leads to a neuronal behavior which is that of an integrator; another type of behavior, closer to that of a resonator, with dampened oscillations is also possible for $E_u < E_0$ and is discussed in section 3.4 and in S1 Text.

In this situation, due to the nonlinearity of the $\epsilon$-nullcline, the biophysically acceptable domain for steady states ($\epsilon \geq 0$ and $V$ in a reasonable range of potential) can contain either zero, one, or three FPs. In the case of a single, necessarily stable FP, it corresponds to a standard neuron with a single resting state. For certain combinations of the neuronal parameters, the $V$-nullcline can intersect the $\epsilon$-nullcline three times, leading to two stable FPs and one unstable point. This situation corresponds to a bistable cell, where two distinct resting states are possible: an up-state, characterized by lower energy levels and high membrane potential, and a down-state, associated to higher energy and hyperpolarized membrane potential. Responses of the bistable neuron to the different step-currents are illustrated in Fig 3. Depending on initial state and the input the neuron transitions between the up- and down-states. Finally, the situation without FPs in the biophysical domain is unsustainable and will lead to

 Simple models including energy and spike constraints to study neuronal disorders

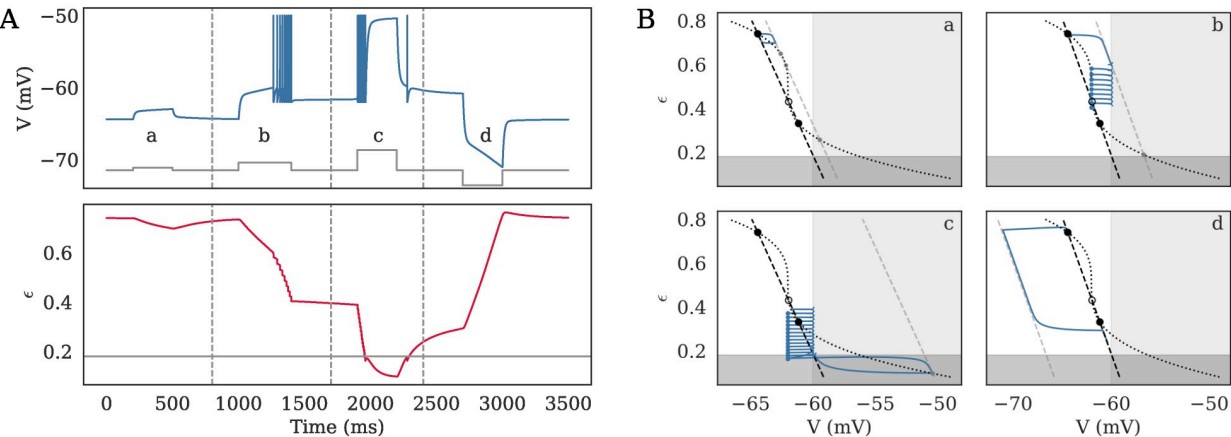

**Fig 3. Dynamics of the _e_LIF model as timeseries (A) and in phase-space (B) in the bistable regime. A**. The behavior of the model is shown in response to four different inputs, shown in grey on the _V_ subplot: a low depolarizing current (a: 10 pA), a stronger depolarizing current (b: 30 pA), a large depolarization (c: 80 pA), and a hyperpolarizing current (d: -60 pA). For visualization purposes, action potentials are made visible by setting the voltage to -50 mV at spike time. **B**. Corresponding behavior in phase-space is shown, each subplot corresponding to one of the four domains separated by the grey dashed lines on panel A. The black curves mark the resting nullclines and the light grey line marks the input-driven _V_-nullcline; resting fixed points (FPs) are marked by the large black circles while input-driven FPs are show by the small grey circles and spike emissions are marked by empty left triangles while reset positions are marked by blue dots. The neuron displays the following behaviors: (a) subthreshold dynamics, where the neuron temporarily leaves the high-energy FP, associated to the down-state, then goes back towards it, (b) transition from the initial high-energy FP to the low-energy FP (up-state) through a spiking period, (c) transition from the up-state to a depolarization block via a spiking period before returning towards the up-state, (d) transition from up- to down-state. See S2 Table for detailed parameters.

rapid neuronal death. Possible reasons for transitions between these states will be detailed in the following section.

We use the transitions in the number of FPs, called _bifurcations_, to predict the behavior of the neuron. The bifurcations can have two separate kinds of consequences, that can potentially happen simultaneously: a) a change in the steady-state behavior of the neuron such as the switch from a unistable to a bistable state or vice-versa, b) a transition from a quiescent to an active state.

Let us discuss these bifurcations in response to an external stimulation associated to an applied current $I_e$. The consequence of $I_e$ is to shift the _V_-nullcline horizontally (towards more negative potentials if $I_e < 0$, or towards more positive if $I_e > 0$), which can lead to transition between the unistable and bistable states as one stable FP either splits into one stable and one unstable FP or, on the contrary, merges with the unstable FP and disappears. This type of transition is called a saddle-node bifurcation and occurs for:

$$I_{e\pm}^* = g_L \left[ E_f - E_u + \alpha(E_u - E_0)\left(1 \pm \frac{2}{3}\sqrt{\frac{\alpha(E_u - E_0)}{3(E_d - E_f)}}\right)\right] \qquad (3)$$

Depending on the value of $I_e$, the neuron can thus display either a single or two stable FPs —see Fig 3 and S1 Text for the analytic derivation of the FPs and S1 Fig for the $I - f$ curves.

As $I_e$ increases, the transition from three FPs to one FP can also lead the neuron to fire, either transiently if the remaining FP is located in the continuous region (if either $V_{FP} < V_{th}$ or $\epsilon_{FP} < \epsilon_c$) or continuously (if $V_{FP} \geq V_{th}$ and $\epsilon_{FP} > \epsilon_c$).

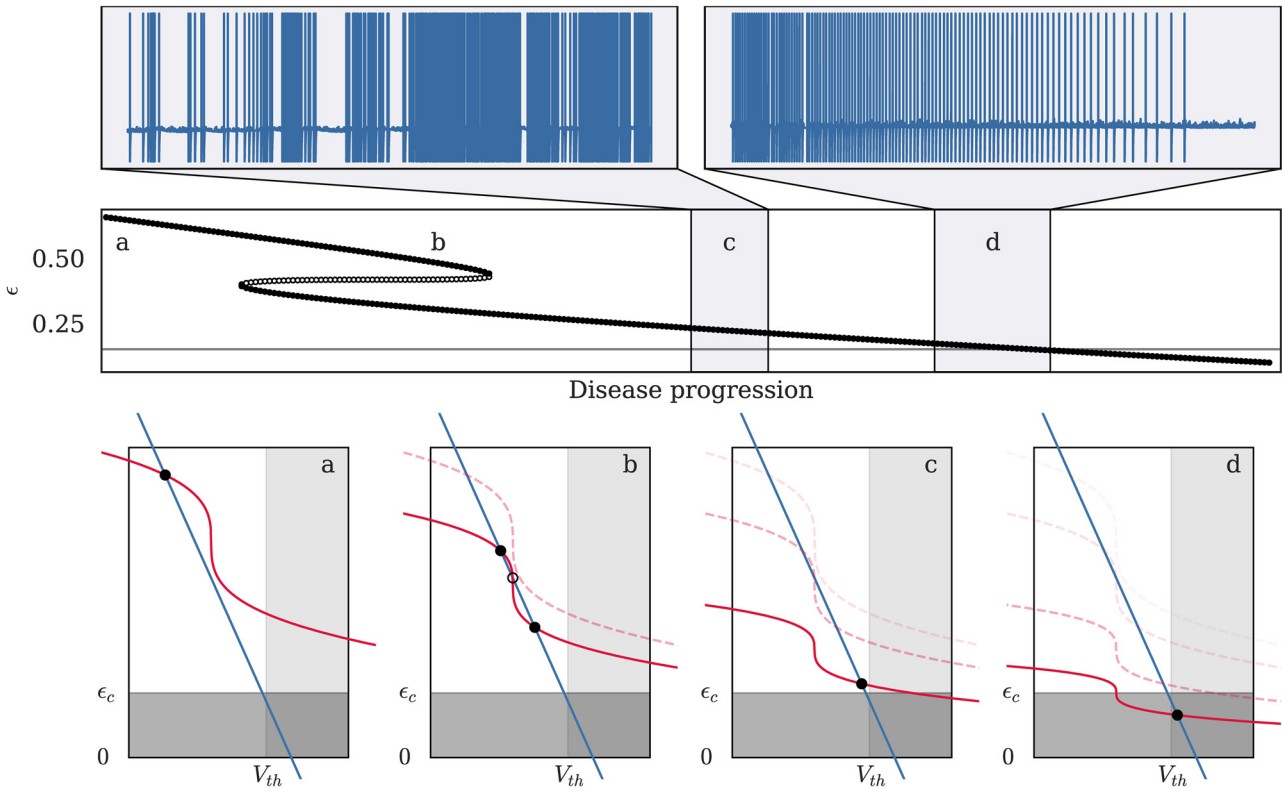

**Fig 4. One possible pathway for the transition between healthy and diseased state in the *e*LIF model.** In the model, progressive decrease in the "energetic health" factor $\alpha$, from 1 to 0.3, leads to a succession of changes in both the number of fixed points (FPs) and in their properties. The middle panel shows the evolution of the FPs' energy levels—filled circles for stable FPS, empty for unstable FPs—with the grey line marking $\epsilon_c$. Four stages of the disease progression are also illustrated in phase-space: (a) healthy neuron with a single FP. (b) bifurcation to a 3 FPs state without major changes in the dynamical properties (susceptible but potentially "asymptomatic" cell). (c) bifurcation to a single low-energy FP associated to an extremely excitable state (diseased cell). (d) further decrease of the energetic health brings the FP below the energy threshold $\epsilon_c$, leading the neuron to become unresponsive. In stages (a) and (b), the neuron lies in its resting state in the absence of input; however, at stages (c) and (d), the two insets on the upper panel show the membrane dynamics of the neuron for a hypothetical "accelerated evolution" of the disease, where the neuron respectively enters (35-second simulation) and leaves (45-second simulation) the "hyperactive" region where usually subthreshold inputs (here modeled by a Poisson noise) are sufficient to trigger uncontrolled spiking. See S2 Table for detailed parameters.

## 3.2 Transition from health to disease

As energy availability decreases, either due to disease [25] or hypoxia [27], neurons often display a parallel increase in their resting membrane potential and excitability, which can lead to highly active periods before the neuron ends up in a highly depolarized yet completely non-responsive state also called depolarization block. Biologically, this low-energy state—(d) and below on Fig 4—would be associated to deregulation of calcium levels and accumulation of oxidizing agents which eventually lead to cell death (occuring when $\alpha$ reaches zero in the model).

Due to the interaction between energy and membrane potential in the *e*LIF neuron, the model can reproduce this kind of dynamics through the evolution of one or more parameters. The most straightforward way to model this transition is through the $\alpha$ parameter which represents the energetic health of the neuron—see Fig 4. The progressive decrease in the value of $\alpha$, from values close to 1 for a healthy neuron to values that tend towards zero for a diseased cell, leads to progressive changes in the membrane potential and excitability of the neuron. The

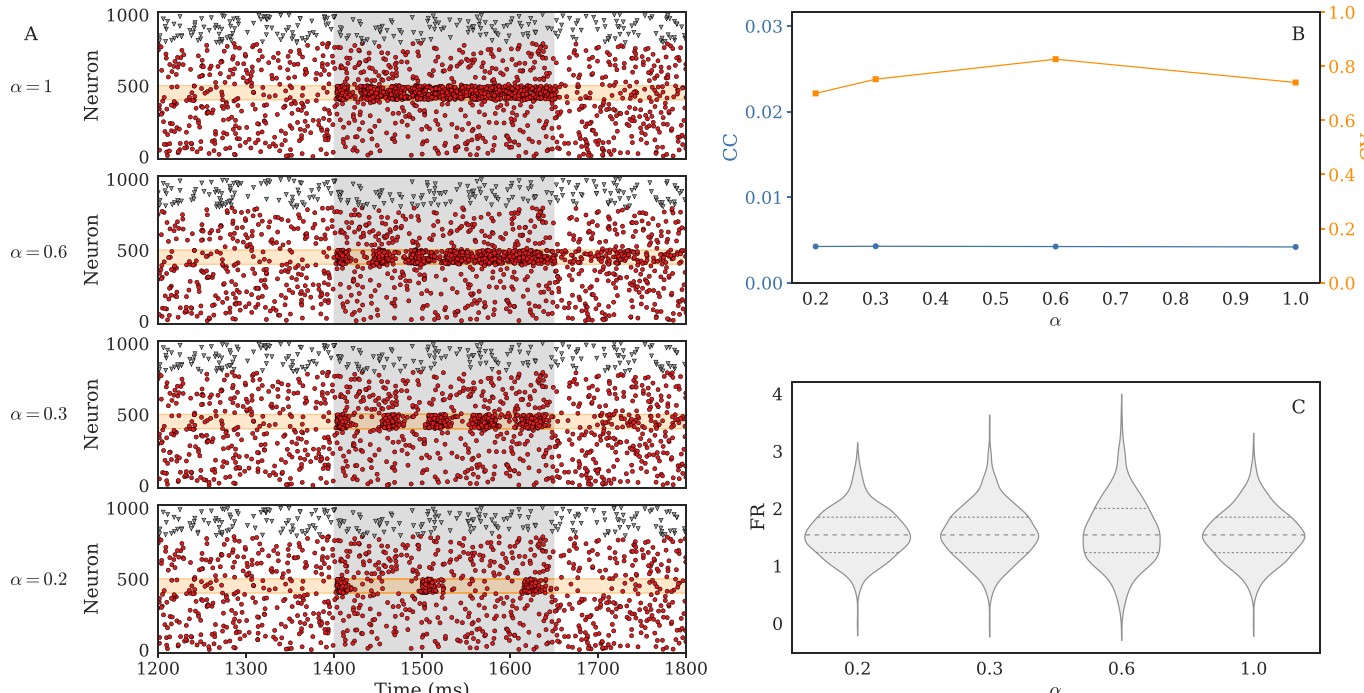

**Fig 5. Decrease in health can be partly compensated by homeostatic mechanisms and be invisible from the statistical properties of background activity, as shown by the behavior on an excitatory and inhibitory population with $N = 1000$ neurons in the asynchronous irregular (AI) state. A**. For such a network, changes in the neuronal health, modeled by a decrease in the $\alpha$ parameter, do not appear in the background activity of the raster (non-grayed areas), where the activity of both excitatory (red circles) and inhibitory (gray triangles) neurons remain very similar. To see the actual consequences of the decrease in health, one must look at the response of the network to an additional input, which is shown in the grayed areas on panel A. In response to a threefold increase in the rate of Poisson input between 1400 and 1650 ms, the activity of 100 excitatory neurons (marked by the orange area) progressively switches from continuous tonic firing (top) to well-separated bursts (bottom). **B**. More quantitative analyses also confirm that the background activity remains close to Poissonian, with coefficients of variation (CVs) around 0.7–0.8, and asynchronous, with an average cross-correlation (CC) smaller than $1/\sqrt{N}$. **C**. The distributions of firing rates over 5 seconds remain almost identical and centered around 2 Hz; dotted and dashed lines respectively denote the quartiles and medians.

typical behavior of the model, illustrated on Fig 4, consists of a slow increase of the resting membrane potential, and thus of the excitability, until the background noise or external input is sufficient to trigger spike emission from the neuron. Once that happens, the cell enters a highly active state in which it remains until the progressive decrease of $\alpha$ brings the target energy below $\epsilon_c$, at which point spike emission stops and the neurons enters a highly depolarized and non-responsive state.

When it comes to collective dynamics, a decrease in neuronal health can go unnoticed especially if the homeostatic regulation adjusts excitability of individual neurons. It happens, for instance, in excitatory and inhibitory networks displaying asynchronous-irregular (AI) activity with low firing rates—see Fig 5.

Without external input, the distribution of firing rates as well as the average properties of the activity (cross-correlations and coefficients of variation) can remain stable despite neuronal health decrease (Fig 5B and 5C). This happens because compensatory mechanisms enable neurons to maintain firing rate homeostasis by means of synaptic scaling and regulation of cell excitability, that we modeled numerically by a decrease of excitatory synaptic weights and an increase of $V_{th}$—see S1 Text and S5 Table for more details.

However, the response to an external input can be drastically modified (Fig 5A), transitioning from an almost continuous tonic response (top), to an intermittent, bursty dynamics (bottom). This example demonstrates how, depending on the homeostatic capabilities of the brain

region of study and the recording protocol, the effect of energetic constraints can be either masked or clearly visible in the neuronal responses. There are multiple ways in which the energetic health can influence the information processing capabilities. Using our models these mechanisms can be studied further in large recurrent networks.

### 3.3 Dynamics of the mAdExp model, biologically-relevant behaviors

Despite the multiple interesting features of the *e*LIF model, several important dynamics such as bursting or adaptation cannot be reproduced within the model. In order to recover all relevant behaviors, we added a spike-generation mechanism as well as an adaptation current to the *e*LIF model to obtain the mAdExp model (modified AdExp with energy dependency).

This 3-dimensional model is then able to provide all the features of the *e*LIF and AdExp models while bringing the dynamics closer to biological observations, especially in large-input or stress-inducing situations. Fig 6 shows several standard neuronal responses reproduced by

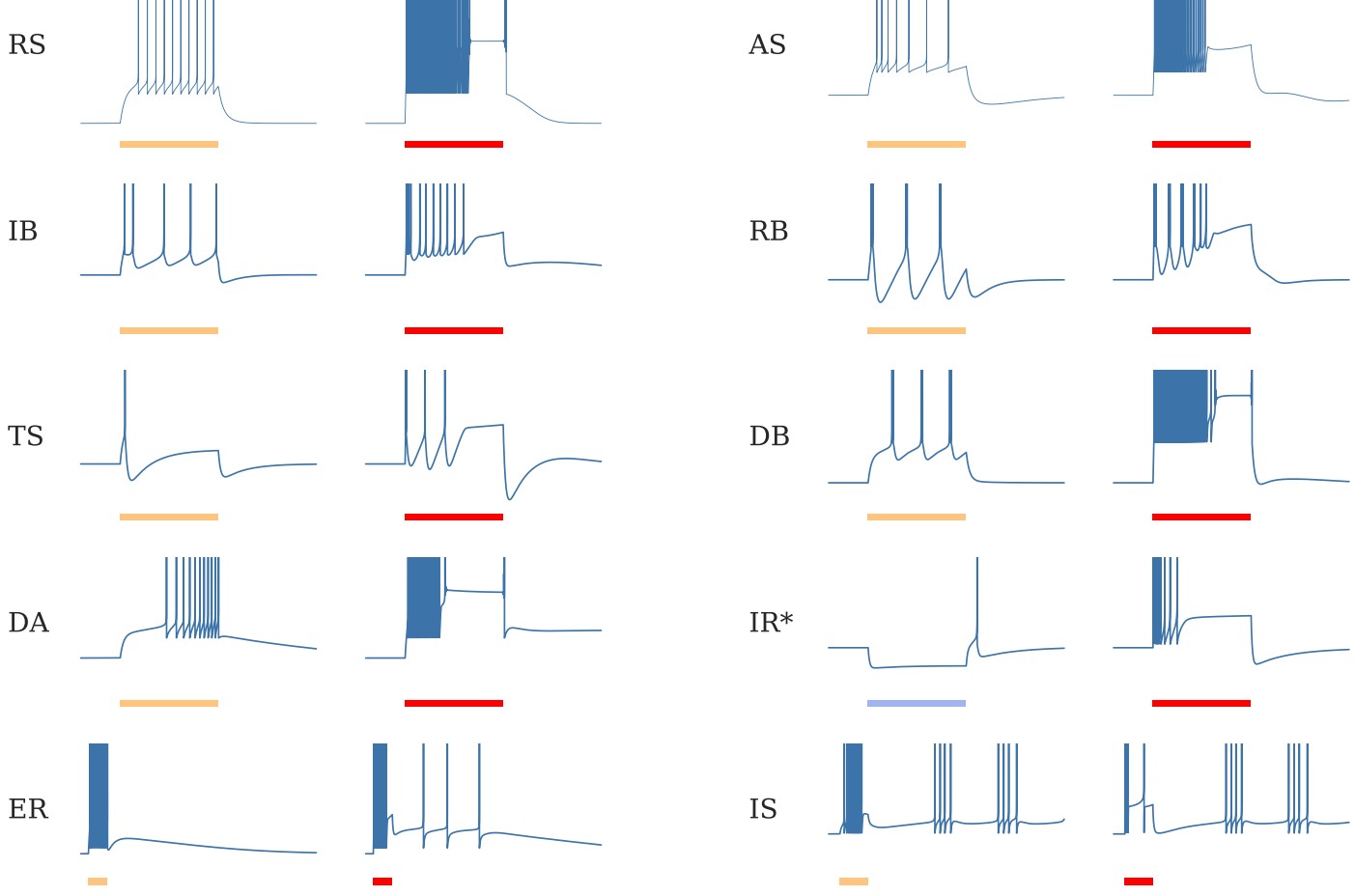

**Fig 6. Typical dynamics of the mAdExp model with different parameter settings in response to current steps given by the scale bars—500 ms for all entries—In yellow to mark lower excitation, red to mark higher excitation, blue bar and asterisk on IR to mark inhibitory current.** The behaviors include regular spiking (**RS**), adaptive spiking (**AS**), initial burst (**IB**), regular bursting (**RB**), transient spiking (**TS**), delayed bursting (**DB**), and delayed accelerating (**DA**). Similar responses to the lower (yellow) currents can be achieved by the original AdExp model. However, each of these dynamics now comes with an "energy-depleted" state for high input current (red), associated to a depolarization block (responses associated to red bars), that cannot be captured by AdExp model. In addition to these standard behaviors, dynamical repertoire of the mAdExp neuron also includes a different mechanism for post-inhibitory rebound spiking (**IR**), and can display post-excitatory rebound (**ER**) or intermittent spiking dynamics (**IS**). See S3 Table for detailed parameters.

the model, as well as how these responses evolve as the input intensity increases up to values where the neuron cannot sustain continuous activity.

Though the theoretical analysis of the model becomes more complex, "standard" resting states—meaning that $V_{FP}$ is several $\Delta_T$ smaller than $V_{th} - \Delta_T \ln\left(\frac{E_u - E_0}{\Delta_T}\right)$—for healthy neurons can be very well approximated by the fixed point of the *e*LIF model because the adaptation current is usually close to zero at rest. Furthermore, their response to low-intensity stimuli can be accurately predicted by the AdExp model with the same common parameters and the corresponding $E_L$ value—see S1 Text for detailed calculations. Most healthy neurons thus share the bifurcations associated to the AdExp model [36, 48], with the notable addition of a new bifurcation for rebound spiking which will be developed in the next section.

## 3.4 Rebound spiking mechanisms in the different models

Rebound spiking is a common property in neurons, with is potentially significant in epilepsy [49] and for information processing, be it in the striatum [50], the thalamocortical loop [51], or in auditory processing [52] and grid cells response generation [53, 54].

This mechanism, though already available in several models such as AdExp [35], strongly restricts the responses of the neuron such that only a fraction of the typical dynamics of rebound-spiking neurons can be recovered. The reason is that, in the AdExp model, rebound bursting is always associated to a sag and significant adaptation—see conditions in [48] and S1 Text—and therefore cannot reproduce either non-sag subthreshold responses or some spiking behaviors associated to excitatory inputs, cf. Fig 7.

The mAdExp model provides two new ways of extending the variety of rebound behaviors that can be modeled: a) by introducing a new mechanism for rebound spike generation without inhibitory sag and b) through the energy dynamics, leading to less significant sags and lower excitability compared to the adaptation mechanism—see also S2 Fig.

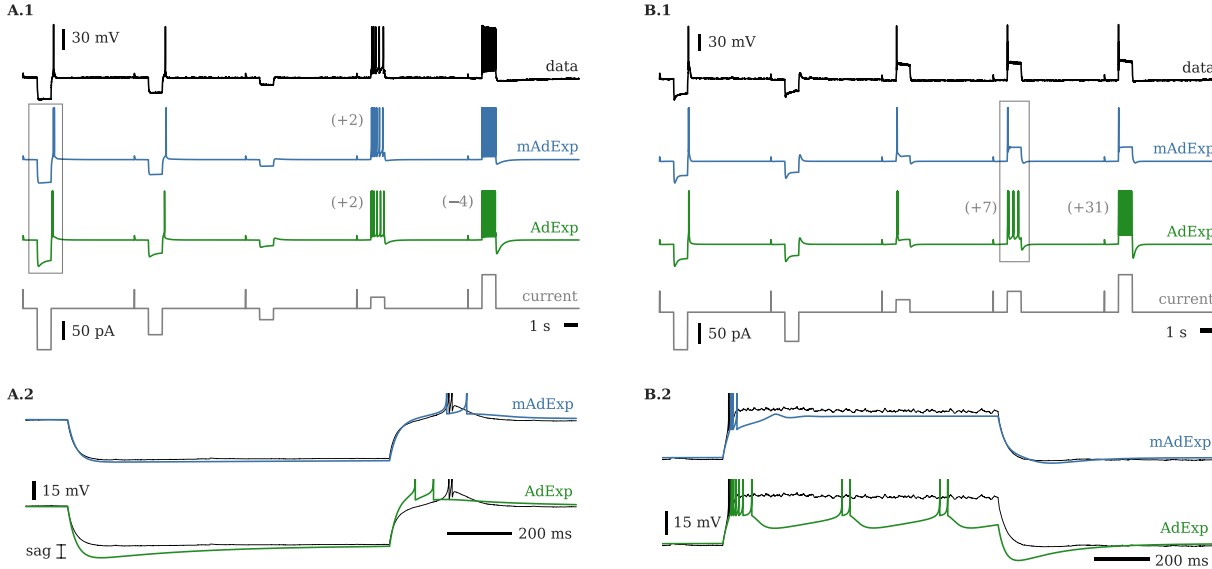

**Fig 7. Voltage traces for two cell types (566978098 and 570896413 in Allen Brain Atlas) and associated fits with mAdExp and AdExp neuron models.** Fourth row represents the input current. Additional or missed spikes are marked in parentheses on the left of the associated spike train. Activities in the rectangles are expanded in the lower panels. **A.** Cell presenting little to no sag upon hyperpolarization and adaptive spiking behavior (A.1); expanded activity (A.2) enables to see the discrepancies between the AdExp model (green) and the data (thin black line) while mAdExp (blue) matches the dynamics much more precisely. **B.** Cell presenting significant sag upon hyperpolarization and almost immediate depolarization block upon depolarizing input (B.1). Both AdExp and mAdExp match the rebound dynamics; however, AdExp cannot reproduce the depolarization block as shown in the expanded dynamics (B.2). See 5 for detailed parameters.

Rebound spiking in mAdExp can occur through a new bifurcation for $Eu - E_0 < \Delta_T$ and $V_{th}$ sufficiently low (see S1 Text for details) which leads to the positive divergence of the $V$-nullcline before $V_{th}$ and thus to the existence of a stable fixed point such that $V_{th} > V_{FP} > V^*$, with

$$V^* = V_{th} + \Delta_T \ln \left( \frac{E_u - E_0}{\Delta_T} \right) < V_{th}.$$

Fig 7 shows how the mAdExp model can successfully reproduce complex behaviors found in the Allen Cell Types Database (available from: celltypes.brain-map.org) such as rebound bursting with little to no sag (cell ID 566978098 shown in Fig 7A.2) or cells displaying both rebound spiking and rapid depolarization block (cell ID 570896413 shown on Fig 7B.2). Due to the mAdExp properties, the possibility of rebound dynamics is thus extended compared to the AdExp model and can be obtained with or without sag, as well as with or without spike adaptation.

## 4 Discussion

### 4.1 Choices underlying the models

The *e*LIF and mAdExp models where chosen as integrate-and-fire models because of the analytic simplicity of such equations and their computational efficiency compared to conductance-based models. Indeed, the straightforward detection of spike times in such models makes them especially suited for simulations of large scale neuronal networks using standard spike-based simulators and their discontinuous dynamics makes bursting possible with only two equations instead of three for continuous models like the Hindmarsh–Rose neuron [55] or general conductance-based models.

Though our models are almost completely phenomenological, their parameters can be directly related to biological phenomena, often even in a quantitative manner, enabling precise predictions from their theoretical analysis. The objective of obtaining single-neuron models where the variables can be interpreted and mapped in a straightforward manner also prevented us from working with previous models such as the Epileptors [56, 57] or Model 2 from [26]—see discussion below.

To obtain a model capable of reproducing all the behaviors that we deemed necessary, the mAdExp model was derived from the AdExp neuron [35] and not from other well known implementations such as the QIF, first proposed by Izhikevich in his seminal paper [34]. This choice was made because, despite some obvious drawbacks regarding the more complex analytics and slightly slower integration of its exponential term, the AdExp model best reproduces the I-V curve of neurons and the dynamics of spike initiation [58], and is exempt of some of the mathematical shortcomings of the QIF model [59].

The $\epsilon$ variable was designed to qualitatively reproduce biological mechanisms and behaviors associated to neuronal metabolism. However, the complexity of these mechanisms led us to choose a strongly reductionist approach to reproduce some of the features that came out of previous studies using more detailed conductance-based or multicompartmental models [9, 25, 26, 60]. Therefore, though it can be qualitatively mapped to some specific mechanisms, $\epsilon$ cannot be quantitatively related to any biological measurement.

### 4.2 Novelty and biological relevance

The *e*LIF and mAdExp neurons are the first integrate-and-fire models to provide an unambiguous description of energy dynamics, enabling to investigate its consequences in single-cell or

in recurrent-network configurations. Indeed, contrary to previous models where slow variables were usually introduced to model adaptation from calcium-gated potassium and bursting—this is notably the case for the z variable in the Hindmarsh–Rose model, the u and w variables in the Izhikevich and AdExp models, or adaptive-threshold models—the $e$LIF and mAdExp models provide the $\epsilon$ variable as a way to explicitly model energy-related and spike-initiation constraints. Though other implementations of models including slow variables might be able to reproduce some of the behaviors examined here, to the best of our knowledge, the $e$LIF and mAdExp are the only phenomenological models that permit the investigation of feedback loops between energy levels, neuronal excitability, and spike emission. The fact that the model's variable are interpretable and directly linked to biological mechanisms enabled us to extend the AdExp model in a straightforward manner; this will also let others expand the models if they need to capture additional mechanisms or external interactions, e.g. with glial cells.

The only other examples of models with an explicit variable representing energy levels we found were developed in [61] and in Model 2 from [26]. While the former is quite simple and not connected to any specific biological mechanism, the latter explicitly presents the second variable as a proxy for the ATP concentration in the neuron. In Model 2, the interpretation of $A$ as a proxy for ATP level notably stems from the fact that it was designed as a simplification of the more detailed Model 1, a conductance-based model that included K-ATP channels, where the $A$ variable was defined as the ATP concentration. The model provides interesting dynamical properties and enabled the authors to develop a new way of modeling the neuro-glio-vascular system. However, if one's purpose is to investigate dynamics where both calcium-gated potassium adaptation and energetic constraints are involved, then one would not be able to use Model 2 as it does not provide a clear distinction between these two mechanisms. Indeed, the newly introduced $A$ variable only influences the value of the threshold and is therefore quite close to a GLIF model [62], which makes it impossible to separate effects that would biologically stem from "standard" adaptation mechanisms, associated to calcium-gated potassium currents, and effects that would be specific to ATP-related dynamics.

The implementation chosen in the mAdExp model solves this issue by establishing a clear separation between the retroactions associated to adaptation and those related to changes in energy levels. In this model, the effects of $\epsilon$ and $w$ on the membrane potential can be opposite and occur (in general) on different timescales. The $\epsilon$ variable also regulates the spike initiation mechanism of the neuron, meaning that, contrary to previous models, energy depletion may render a neuron totally unresponsive regardless of the input strength. In addition, our models consider all sources of energy consumption including spikes and subthreshold ion currents.

Thus, over long timescales, the $\epsilon$ parameter qualitatively accounts for energy availability as the ATP/ADP ratio to which pumps and channels are sensitive [39, 63], with this sensitivity summarized in the $I_{KATP}$ parameter of the mAdExp model. Contrary to slow current variables that can vary arbitrarily into the positive and negative realm, $\epsilon$ represents an energy stock that must remain positive for the neuron to survive: if the neurons encounter conditions where $\epsilon$ reaches zero with $V \geq V_d$, the models provide an explicit condition for neuronal death.

Over shorter timescales, sharp decreases in $\epsilon$ following spike emission can lead to depolarization block. This phenomenon is mostly associated with sodium channel inactivation in neurons, and is caused by a sodium accumulation that is too quick to be compensated by the Na/K pump. Though it is not directly related to energetic constraints, we consider that having this mechanism associated to the "energy" variable makes sense because the timescale of sodium channel inactivation depends on the resting sodium levels, which in turn depend on the energetic health of the neuron: a neuron with a very active pump would be able to sustain more

spikes than one with a defective pump. This mechanism is related to the $\delta$ and $\epsilon_c$ parameters in the models.

Finally, as $\epsilon$ represents the ATP/ADP ratio, the $\alpha$ parameter quantifies the metabolic and mitochondrial state of the neuron and can be used to investigate the transition from health to disease as exemplified in the Results. A decrease of the $\alpha$ parameter can be related to metabolic insults associated to either mitochondrial defects [64, 65], a decrease of oxygen or glucose availability, or the buildup of various molecules such as reactive oxygen species (ROS) [66, 67] that prevent proper metabolic homeostasis.

### 4.3 Consequences of the V/$\epsilon$ relationship

One of the major features of the model is the interaction between the energy level and the resting potential of the neuron. This interaction can lead to a transition from "healthy" or "optimally responsive" neurons to "diseased", non-responsive neurons. Interestingly the neuron may go through a hyper-excitable state during this transition, meaning that disease progression can be marked by a broad range of neuronal dynamics and properties.

Because changes in the energy level affect the neuronal excitability, the synchronizability and information processing properties of the neurons change significantly as their available energy decreases. This property of the model matches observations in various neurodegenerative diseases. Synchronizability notably changes in Parkinson's disease (PD), for instance, where oscillations in the beta range (13–30 Hz) become predominant and are thought to be involved in some motor symptoms. Though known variations in the connectivity strongly influence this dynamical change, modification of intrinsic neuronal properties due to metabolic insult are also likely to contribute to the transition towards more synchronized activity [32, 68]. Even more obvious, epileptic seizure are characterized by excessive or hypersynchronous neuronal activity and their onset and termination are likely to be related to the metabolic state of the neurons [19, 20, 69]. Finally, the transition through an hyperactive phase before entering the non-responsive depolarized state has also be proposed for diseases such as ALS [25].

From an information transfer perspective, the positive retroaction between depolarization and energy depletion can lead to increased false positives due to hyperexcitable neurons in diseased conditions. Furthermore, because of the necessity of a minimum "metabolic level" for spike emission, this also means that energy-impaired neurons cannot sustain long-term responses, and would tend to display phasic responses. These combined effects could further drive bursty activity such as what is observed in PD, where the reliability of thalamic relay breaks down and the cells start emitting bursts of activity which could lead to tremor [70, 71].

The mAdExp model can reproduce the main relevant dynamical properties in these phenomena and therefore enables detailed and potentially large-scale computational studies. Such simulations could lead to more realistic dynamical models and thus to new experimentally testable predictions.

### 4.4 Limitations

Due to their simplicity, the *e*LIF and mAdExp models still suffer from many of the limitations of the original LIF and AdExp models.

For example, the *e*LIF cannot reproduce bursting behavior and can only exhibit simple accelerating or decelerating spiking patterns. Though the dynamical richness of mAdExp is greater than the LIF and AdExp models, its adaptation mechanism also possesses the same drawbacks as the original model: the presence of a single adaptation timescale $\tau_w$. As for the

AdExp and QIF, though, the mAdExp can be extended in a straightforward manner to account for multiple adaptation timescales by adding additional $w$-like variables.

Since multiple biological phenomena are associated to or can affect the $\epsilon$ variable (Na$_f$ inactivation, ATP/ADP ratio, oxygen concentration, pH, ROS. . .), precise experimental verifications and relations to biochemical pathways can be quite complex or even impossible to predict, at least if several phenomena are occurring on similar timescales. The depolarization block, for instance, only stems from the combination of metabolic parameters $\delta$ and $\epsilon_c$ in our model. It is indeed strongly related to sodium or potassium accumulation (though not 'metabolic' per se, these directly depend on the efficiency of the NKP), to general energetic considerations [72, 73], and may have been selected due to energetic constraints [74]. Yet, other slow mechanisms that are not accounted for in our models also contribute to this behavior in biological neurons; notably chloride-related changes in cell volume [75]. However, the purpose of our models is to explore how changes in energy availability may increase or reduce the occurrence of specific behaviors such as the depolarization block. Thus, for the sake of simplicity, we did not include such passive mechanisms as they do not directly influence energy availability.

Eventually, complex interactions between sodium or calcium levels and ATP production [76, 77] is only coarsely implemented in the model. In particular, because the adaptation variable $w$ represents calcium-gated potassium, and not directly the calcium levels, interactions between $\epsilon$ and $w$ would not capture precise biological mechanisms. Overall, calcium dynamics can have very different impacts on ATP production, depending on concentrations and timescales, which cannot be completely accounted for by the simple relationship present in the model.

## 5 Conclusion

The two models introduced in the present study provide a novel reductionist approach to include generic energetic constraints and energy-mediated dynamics to the models of single neurons. The low-dimensional nature of these two dynamical systems makes them suitable for analytical investigation of energy-based bifurcations in neuronal behaviors, as well as for large scale simulations.

The mAdExp model, in particular, is able to replicate a large range of biologically-relevant behaviors as well as their evolution under metabolic stress. Complex behaviors that are crucial for some brain regions and disorders, such as rebound spiking or depolarization block, now can be successfully reproduced. Since energetics plays a critical role in many disorders, this model is especially well suited to explore possible origins of the differences observed between normal and diseased activities in neuronal populations.

Finally, these new models are not limited to the comparison between specific healthy or diseased states, as they provide a tunable parameter to represent neuronal health. Thus, the continuous transition between states can now be investigated, as well as dynamical feedback between activity and resource consumption in resource-limited conditions such as in neuronal cultures or seizures.

## Supporting information

**S1 Text. Supporting information for "Simple models including energy and spike constraints reproduce complex activity patterns and metabolic disruptions" including in-depth mathematical analysis, benchmarks, and parameter data.**
(PDF)

**S1 Fig. $I - f$ curves of the $e$LIF neuron for different threshold values $V_{th}$ (left/right).** The corresponding phase-space is shown in the middle. Threshold values are -65.5 (dark grey), -63,

-61, and -59 mV (light grey); they correspond to the associated curves on the $I - f$ plots and to the dashed vertical lines on the phase-space representation. The type of the curve depends on the position of $V_{th}$ compared to the position of the low-energy fixed point (FP) at the bifurcation point which is shown as a filled black circle: for $V_{th} > V_{FP}$, the neuron has a continuous type I response curve whereas for $V_{th} > V_{FP}$ the curve, though still continuous, becomes closer to a type II curve, with a sharp increase starting immediately at the bifurcation current $I_e^*$. See S2 Table for detailed parameters.
(EPS)

**S2 Fig. Dynamics of the *e*LIF model as timeseries (left) and in phase-space (right) for $E_u < E_0$ (resonant behavior).** The behavior of the model is shown in response to four different inputs, shown in grey on the $V$ subplot: a low depolarizing current (a: 10 pA), a stronger depolarizing current (b: 30 pA), a large depolarization (c: 80 pA), and a hyperpolarizing current (d: -60 pA). Corresponding behavior in phase-space is shown in the four right panels, with spike emission marked by an empty left triangle and reset position marked by a dot: (a) the neuron leaves the fixed point (FP), then goes back towards it (both transitions are associated to and up/downshoot), (b) the neuron spikes at decreasing frequency as its energy is depleted, (c) the neuron spikes, then enters a depolarization block for high stimulation, (d) post-inhibitory overshoot is associated to rebound spiking. See S2 Table for detailed parameters.
(EPS)

**S1 Table. Runtime of various models in NEST.** A "baseline" run with no neuron (None), compared to runs with one neuron of each of the mentioned models. For the new energy-based models (*e*LIF and mAdExp), two runs were performed: one using a naive implementation and another using slightly optimized implementation (numbers in parentheses). Conductance-based models are also included: a standard Hodgkin-Huxley (HH) model which can display regular spiking an depolarization block, and one with calcium and calcium-gated potassium (HH+Ca) to reproduce bursting dynamics.
(PDF)

**S2 Table. Parameters used with the *e*LIF model.**
(PDF)

**S3 Table. Parameters used for the different behaviors of the mAdExp model on Fig 6.**
(PDF)

**S4 Table. Parameters used to match rebound spiking behaviors on Fig 7.**
(PDF)

**S5 Table.** Left: static neuronal parameters used with the all *e*LIF neurons in Fig 5. Right: Specific parameters used for each of the simulations at a different neuronal health in Fig 5. Each health level, corresponding to a value of $\alpha$ is associated to the corresponding values for $V_{th}$ and $V_{reset}$ in the same column.
(PDF)

## Author Contributions

**Conceptualization:** Tanguy Fardet, Anna Levina.

**Formal analysis:** Tanguy Fardet.

**Funding acquisition:** Tanguy Fardet, Anna Levina.

**Investigation:** Tanguy Fardet.

**Methodology:** Tanguy Fardet, Anna Levina.

**Project administration:** Tanguy Fardet, Anna Levina.

**Resources:** Anna Levina.

**Software:** Tanguy Fardet.

**Validation:** Tanguy Fardet.

**Visualization:** Tanguy Fardet.

**Writing – original draft:** Tanguy Fardet.

**Writing – review & editing:** Tanguy Fardet, Anna Levina.

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
