## [Decision Letter · Decision Letter 0]

2 Jun 2020

Dear Dr. Fardet,

Thank you very much for submitting your manuscript "Simple models including energy and spike constraints reproduce complex activity patterns and metabolic disruptions" for consideration at PLOS Computational Biology.

As with all papers reviewed by the journal, your manuscript was reviewed by members of the editorial board and by several independent reviewers. In light of the reviews (below this email), we would like to invite the resubmission of a significantly-revised version that takes into account the reviewers' comments.

We cannot make any decision about publication until we have seen the revised manuscript and your response to the reviewers' comments. Your revised manuscript is also likely to be sent to reviewers for further evaluation.

Sincerely,

Arvind Kumar, Ph.D.

Guest Editor

PLOS Computational Biology

Kim Blackwell

Deputy Editor

PLOS Computational Biology

Reviewer's Responses to Questions

**Comments to the Authors:**

Reviewer #1: In this manuscript entitled “simple models including energy and spike constraints reproduce complex activity patterns and metabolic disruptions”, Fardet and Levina explore how energy constraints shape neuronal activity. This topic is an important one that deserves increased attention. That said, I do not think this paper delivers on the goals outlined in the Abstract and Introduction. The authors introduce a slow variable into simple models, namely the LIF and AdExp models, and explore the dynamical phenomena that emerge. The simulations and phase-plane analysis are clearly presented, but I have reservations about the biological interpretations and novelty.

Major concerns:

-The authors claim (first sentence of Abstract) to introduce new models that “account for energy supply and demand”. In reality, they reproduce phenomena like depolarization block and bistability/ bursting by introducing a slow variable whose connection to energy is not so clear. These phenomena have been previously shown to emerge from, amongst other things, changes in ion reversal potentials that occur if/when ion flux is tracked AND when the resulting ion concentration changes are not counteracted by pumps or co-transporters that (directly or indirectly) rely on ATP consumption. The dynamics introduced by slow ion concentration changes have actually been the subject of significant modeling, which is not adequately discussed in this manuscript. Those past papers may not have referred to the slow variable as “energy”, but there is certainly a connection. The novelty of the current paper is much reduced once one recognizes this connect.

-Despite providing a reasonable introduction about the importance of energy for neuron function, the modeling failed to connect energy consumption and neuronal excitability in a meaningful way. According to equations, E_L is dependent epsilon (energy) but the relationship is not explained. Why is the generation of energy written the way it is? For the mAdExp model, w also depends on I_katp, but, here again, the relationship is not explained and one is left wondering if parameters have any biological meaning/constraint, or are simply adjusted to make the model work. On lines 312-313, the authors write that “epsilon cannot be quantitatively related to biological measurement”, but even the qualitative mapping falls short.

Granted, the authors want to create a simple model amenable to analysis and large-scale simulations, but in justifying the models they propose, they might have created a more biophysically detailed model in which energy-dependent mechanisms are explicitly modeled, and then formally reduced it to the forms presented here… or they might have derived their equations from known relationships that they can point to in the literature. Without this, the modeling seems circular – models produce the “behavioral requirements” they were designed to reproduce – and one is left wondering if that is really because of energy, and if the models can do more than they were designed for. Indeed, the authors speculate about synchrony and information processing, but test neither; based on their mention of large-scale simulations in the Abstract, I was expecting more.

Reviewer #2: Simple models including energy and spike constraints

reproduce complex activity patterns and metabolic

disruptions

In this computational study, the authors present simplified neuron models in which neural dynamics is coupled to the dynamics of an abstract energy variable. This kind of coupling is worked out for two standard spiking neuron models – Leaky Integrate and Fire (LIF) model and AdExp model. It is an important line of work since there is growing evidence that a lot of neurological disorders have their roots in impaired metabolism. It is essential develop simple neuron models that explore energy-dependence of neural firing so as to better understand metabolic underpinnings of neural disease.

Major Concerns:

I have some objections regarding the “energy” aspect of the proposed models.

The proposed models have a neural component and an energy component. For the former, the authors used standard spiking neuron models which have been proven to explain real neural firing patterns under certain conditions. That is, their link to real neural spiking was studied extensively.

But in case of energy component of the proposed model, no effort seems to be made to link it to real energy substrates in the brain like ATP, pyruvate, lactate glucose etc. In the conceptual schematic of fig 1, the authors state that the energy ATP/ADP etc but it only remains a general comment. It does not reflect in the mathematical development of the equations.

Here again, in eLIF model, the energy dynamics is completely abstract and the authors do not even comment on its relation to real metabolic substrates of energy in the brain.

In eAdExp model, in the w-dynamic equation, there is a I_KATP term where the model makes its first contact with its biophysical substratum. But, unfortunately, the authors don’t define I_KATP term and only make an unsubstantiated comment that I_KATP is used…

“to account for ATP-sensitive potassium channels that trigger potassium outflow when the ATP/ADP ratio becomes small.”

The authors need to explain this comment in much greater detail and show how the proposed energy dynamics has a deeper biophysical grounding.

In this respect, the authors must refer to the work of (Chhabria and Chakravarthy 2016). In this study, the authors describe spiking neuron models with an associated dynamics of an energy variable, analyse different dynamical regimes and link them to normal and energy-starved conditions. The model formulation in (Chhabria and Chakravarthy 2016) has a strong similarity to the proposed models in the current manuscript, which does not cite the previous study.

The similarity is even more interesting if you consider that (Chhabria and Chakravarthy 2016) also used I_KATP current to make contact with real biophysical “energy.”

Therefore, the authors of the current must make a detailed comparison of their models with those mentioned above and justify the novelty, if any, of their proposed model.

Minor comments:

1. In Fig. 3, in the subplots on the right side, in the subplot b, the voltage fluctuation seems to be confined roughly to [-65 mV, -60 mV] but in the corresponding voltage plot on the left top, voltage seems to fluctuate between roughly -60 mV and -50 mV. How do you explain this discrepancy? Same thing is true with subplot c.

2. In fig 4, where dynamics under energy-starved or pathological conditions are described, whether the phase plot has 3 FPs or 1 FP depends on the value of I_e also and not just alpha. So does the change in the input current Ie also represent pathology? Which doesn’t make much sense. Can the authors explain this?

Typos:

On line 219: “bifurcations” instead of “bifurcation”

On line 234: “ends” instead of “end”

On line 305: “shortcomings” instead of “shortcoming”

The authors may scan the whole manuscript to fix all the typos.

Reviewer #3: The dynamics of neurons depends on many physiological parameters. The authors provide two phenomenological single-cell models that incorporate the effect of the metabolic supply – a variable that is particularly relevant in situations of stress and pathology.

They equip the standard LIF model with energy dynamics, including contributions of the consumption arising from subthreshold membrane potential changes as well as action potentials on a phenomenological level. In addition, they also propose an adaptive version of an exponential integrate-and-fire model with an additional dimension and a representation of a sodium-potassium pump current. The authors argue that this approach captures a wide variety of spiking dynamics (including rebound spikes without inhibitory sag) and also takes into account the impact of the energetic supplies. In contrast to the usual phenomenological threshold models, these energetically extended models can also capture the dynamical state of depolarization block.

The idea of combining simple phenomenological models with an energetic component is excellent and paves the road towards larger-scale network simulations and analyses that incorporate the metabolic state. Existing data across qualitatively different dynamics can be captured by the models (the authors provide examples) and the models also allow one to investigate the effects of (usually slow) changes in metabolic supply and the transitions between healthy (energy-rich) and pathological (energy-deprived) states.

The manuscript is well written and the analyses are described clearly. I do not hesitate to give an immediate recommendation to accept the manuscript without major revision. I am convinced that it provides an interesting novel contribution that will meet the interest of the community. The authors may have considered to end their analysis with a network simulation, providing an example for the usefulness of the models in the larger network context. But I do not consider this a requirement.

**Have all data underlying the figures and results presented in the manuscript been provided?**

Reviewer #1: Yes

Reviewer #2: Yes

Reviewer #3: Yes

PLOS authors have the option to publish the peer review history of their article (what does this mean?). If published, this will include your full peer review and any attached files.

Reviewer #1: No

Reviewer #2: Yes: Srinivasa Chakravarthy

Reviewer #3: No
---

## [Decision Letter · Decision Letter 1]

31 Aug 2020

Dear Dr. Fardet,

Thank you very much for submitting your manuscript "Simple models including energy and spike constraints reproduce complex activity patterns and metabolic disruptions" for consideration at PLOS Computational Biology.

Sorry for the delay in the decision. 

As with all papers reviewed by the journal, your manuscript was reviewed by members of the editorial board and by several independent reviewers. In light of the reviews (below this email), we would like to invite the resubmission of a significantly-revised version that takes into account the reviewers' comments.

As you will see two of the three reviewers have endorsed the manuscript for publication. But one of the reviewers has still some outstanding concerns. And as I see it, those are not trivial concerns and without addressing them we cannot proceed. Therefore I would like to invite your revise the manuscript one final time. But please make sure that this time you address those concerns adequately.

We cannot make any decision about publication until we have seen the revised manuscript and your response to the reviewers' comments. Your revised manuscript is also likely to be sent to reviewers for further evaluation.

Sincerely,

Arvind Kumar, Ph.D.

Guest Editor

PLOS Computational Biology

Kim Blackwell

Deputy Editor

PLOS Computational Biology

Dear Dr. Fardet

Sorry for the delay in the decision. As you will see two of the three reviewers have endorsed the manuscript for publication. But one of the reviewers has still some outstanding concerns. And as I see it, those are not trivial concerns and without addressing them we cannot proceed. Therefore I would like to invite your revise the manuscript one final time. But please make sure that this time you address those concerns adequately.

Reviewer's Responses to Questions

**Comments to the Authors:**

Reviewer #1: The manuscript has been improved by the latest revisions, but most of my concerns were not adequately addressed, as explained below.

1. The authors responded to my first point that “slow variables were usually introduced to model adaptation using calcium-gated potassium and bursting dynamics”. But there are a lot more ions to consider that just calcium, and (relatively slow) changes in those ion concentrations have been show to introduce slow dynamics into neuron models (directly by changing reversal potentials, which in turn alters the strength of different currents, amongst other effects). A quick Pubmed search turns up dozens of such models. I list just a few to make sure the authors understand what I am referring to:

https://pubmed.ncbi.nlm.nih.gov/30180647/

https://pubmed.ncbi.nlm.nih.gov/22058273/

https://pubmed.ncbi.nlm.nih.gov/10899222/

I think my original point was pretty clear. I agree that adaptation has been extensively modeled as a calcium-dependent process, but the authors need to consider other slow processes that impact neuronal spiking in ways directly relevant for this study.

2. The authors have revised the text in the Methods section to help connect energy consumption and neuronal excitability, but the outcome fell short. On line 151-153, the authors explain that since “there is no simple relationship between membrane potential and energy consumption, we decided to use a 3rd order polynomial to give a nonlinear shape to the epsilon-nullcline.” Sorry, but I really don’t think that is a legitimate explanation. Why does the nullcline have that shape? The next sentence (because “this enables both quasi-linear behaviors and more complex dynamics”) is circular – we do it because it works. I am still trying to figure out a sigmoidal relationship between ATP/ADP ratio and voltage would arise.

There are other statements, which might seem minor, but raise concerns. For example, the authors suggest that sodium-channel inactivation occurs because “the Na/K pump cannot move sodium out quickly enough” and cite reference 28. Bikson et al. (2003) does not address intracellular sodium concentration; instead, I would interpret their data, which shows dramatic extracellular potassium accumulation in the context of seizure activity, as reducing post-spike repolarization which impairs recovery of Na channels from inactivation. Intracellular accumulation of sodium will reduce the Na reversal potential, and could contribute to depolarization that encourages Na channel inactivation, but intracellular sodium does not directly cause Na channel to inactivate. Such details may not be critical, but the onus is on the authors to argue that, which starts with an accurate description of the biological mechanism followed by a rational simplification for the purposes of modeling. For the record, I do not think that the model has to capture all of the biophysical detail, but I am disturbed when the authors rationalize their model based on misunderstood mechanisms.

3. I appreciate the addition of Figure 5. The legend text refers to red circles and gray triangles, which are not clear in the figure version embedded in the text, but do show up in the larger version of the figure. But in that larger version, shading is black rather than gray. The text says the input rate is increased between 1500 and 1600 ms, but the shaded region (and altered response) seems to extend from before 1400 to after 1600 ms.

Reviewer #2: The authors have addressed the concerns raised by me in the earlier review. I have no further comments and recommend acceptance of the manuscript for publication. However there are still some types, issues with grammar, that need to be addressed.

**Have all data underlying the figures and results presented in the manuscript been provided?**

Reviewer #1: Yes

Reviewer #2: Yes

PLOS authors have the option to publish the peer review history of their article (what does this mean?). If published, this will include your full peer review and any attached files.

Reviewer #1: No

Reviewer #2: **Yes: **V Srinivasa Chakravarthy
---

## [Editor Report · Decision Letter 2]

9 Nov 2020

Dear Dr. Fardet,

We are pleased to inform you that your manuscript 'Simple models including energy and spike constraints reproduce complex activity patterns and metabolic disruptions' has been provisionally accepted for publication in PLOS Computational Biology.

Best regards,

Arvind Kumar, Ph.D.

Guest Editor

PLOS Computational Biology

Kim Blackwell

Deputy Editor

PLOS Computational Biology

Thank you for revising the manuscript. I think the current version can be accepted for publication. The discussion with the first reviewer was very interesting and many new aspect of the model became more visible. Therefore I would like to suggest that you chose to make the reviewer comments history available with the published version of the paper. I think that will be very useful to the readers.

---

## [Editor Report · Acceptance letter]

8 Dec 2020

PCOMPBIOL-D-20-00540R2 

Simple models including energy and spike constraints reproduce complex activity patterns and metabolic disruptions

Dear Dr Fardet,

I am pleased to inform you that your manuscript has been formally accepted for publication in PLOS Computational Biology. Your manuscript is now with our production department and you will be notified of the publication date in due course.

With kind regards,

Livia Horvath
